# Biphenarenes, Versatile Synthetic Macrocycles for Supramolecular Chemistry

**DOI:** 10.3390/molecules28114422

**Published:** 2023-05-29

**Authors:** Wenjie Zhang, Wenzhi Yang, Jiong Zhou

**Affiliations:** 1Department of Chemistry, College of Sciences, Northeastern University, Shenyang 110819, China; zhangwenjie@stumail.neu.edu.cn (W.Z.); 20191248@stu.neu.edu.cn (W.Y.); 2Guangdong Provincial Key Laboratory of Functional Supramolecular Coordination Materials and Applications, Jinan University, Guangzhou 510632, China

**Keywords:** biphenarenes, macrocycles, molecular recognition, applications, supramolecular chemistry

## Abstract

The development of supramolecular chemistry has always been accompanied by the innovation of macrocyclic hosts. The synthesis of novel macrocycles with unique structures and functions will bring new development opportunities for supramolecular chemistry. As a new generation of macrocyclic hosts, biphenarenes have customizable cavity sizes and diverse backbones, overcoming the limitation that the cavities of traditionally popular macrocyclic hosts are generally smaller than 10 Å. These features undoubtedly endow biphenarenes with distinguished host–guest properties, which have attracted more and more attention. In this review, the structural characteristics and molecular recognition properties of biphenarenes are summarized. In addition, the applications of biphenarenes in adsorption and separation, drug delivery, fluorescence sensing and other fields are introduced. Hopefully, this review will provide a reference for the study of macrocyclic arenes, especially biphenarenes.

## 1. Introduction

In the middle of the last century, the discovery of crown ethers laid the foundation for the development of synthetic molecules that could engage in non-covalent interactions [1]. Then, the concepts of supramolecular chemistry and host–guest chemistry came into being [2,3,4]. In the development of supramolecular chemistry, the synthesis of novel macrocyclic hosts with special properties is an enduring topic [5,6,7,8]. Macrocyclic hosts, including crown ethers [9,10], cyclodextrins [11,12,13,14,15], calixarenes [16], cucurbiturils [17,18,19], pillararenes [20,21,22,23,24,25,26], coronarenes [27] and oxatubarenes [28], are attractive supramolecular hosts with inherent cavities. Thanks to their excellent host–guest properties, macrocyclic hosts play important roles in the fields of chemistry [29,30,31,32], materials science [33,34,35,36], biology [37,38,39,40,41], etc.

In 2015, Li and co-workers constructed a series of customizable macrocycles named biphenarenes using a modular synthetic strategy [42]. Since then, the host–guest properties, supramolecular assembly behaviors and functional applications of biphenarenes have been widely explored [43,44,45,46]. On the one hand, it is easy to obtain biphenarene derivatives containing alkoxy, hydroxy and anionic/cationic groups due to the easy synthesis and derivatization of biphenarenes. The selective complexation of cationic, anionic and neutral guest molecules with biphenarenes can be achieved in different solvents [47,48,49,50]. On the other hand, biphenarenes with large cavities can be prepared by reasonably adjusting the number of structural units, overcoming the limitation of traditional macrocycles with small cavities (≤10 Å). Customizable cavity sizes make it possible to encapsulate large guests or molecules, which will effectively expand the application range of biphenarenes. The discovery of biphenarenes has greatly enriched the toolbox of synthetic macrocyclic hosts and will also promote the further development of supramolecular chemistry.

To date, there are few reviews on biphenarenes and they mainly focus on the synthesis and structure of biphenarenes [49]. In this review, the structures and molecular recognition properties of biphenarenes are discussed in detail. In addition, applications of biphenarenes in adsorptive separation, drug delivery and fluorescence sensing are summarized. Owing to their interesting structural characteristics and rich host–guest properties, biphenarenes have broad development prospects in the construction of functional materials. It is expected that this review will provide reference for the study of biphenarenes and their functional materials.

## 2. Structures of Biphenarenes

Macrocyclic hosts are useful tools for the research of non-covalent interactions [51,52,53]. Host–guest properties of traditional macrocycles (such as crown ethers, cyclodextrins, calixarenes, cucurbiturils and pillararenes) have been widely studied in the past few decades [54,55,56,57,58]. However, traditional macrocycles typically have cavity sizes of less than 10 Å (Table 1), and it is very challenging to prepare giant macrocycles. This makes them generally suitable for binding small- or medium-sized guests, but it is difficult to accommodate large guest molecules. Increasing the number of structural units is a common way to increase the cavity size of macrocycles. However, the simple addition of structural units tends to distort the structures of macrocycles, and it is difficult to obtain macrocycles with large cavities. This severely limits the further applications of macrocyclic hosts. In addition, for traditional macrocycles, functional groups are usually introduced through their portals rather than the skeleton.

In order to solve these problems, many macrocyclic compounds with large, rigid cavities have been developed. For example, cycloparaphenylenes, consisting of a simple string of benzene, have attracted scientists because of their simple and beautiful structure and potential applications in materials science and supramolecular chemistry [59,60]. In 2015, Li and co-workers synthesized a new macrocyclic host named biphenarenes, which including basic biphen[*n*]arenes, functional biphen[*n*]arenes, and cage compounds [42]. Typically, biphenarenes are made up of 4,4′-biphenol or 4,4′-biphenol ether units linked by methylene bridges at the 3- and 3′-positions. The synthesis of biphenarenes is based on the linking of reaction modules to form macrocycles by Friedel–Crafts alkylation. In addition, modular synthetic strategy is a versatile method for the synthesis of biphenarenes, which can increase the cavity sizes by changing the structural units (Figure 1) [48,49]. For example, the cavity sizes of biphenarenes can be easily increased using long and rigid structural units or increasing the number of structural units. Meanwhile, gram-scale synthesis of biphenarenes is easily achieved in a laboratory. The purification of biphenarenes can be achieved by column chromatography and recrystallization. Furthermore, biphenarenes are easy to prepare since they can be obtained by a one-step condensation reaction using commercial reagents. Biphenarenes show good performance in adsorptive separation, sensing and drug delivery, and have broad application prospects in chemistry, biology, materials science and other fields.

Compared with traditional macrocycles, the structures of biphenarenes have two advantages:

(i) Large cavity. Like many other structurally related macrocyclic hosts, biphenarenes are formed from suitable electron-rich aromatic building blocks and formaldehyde by repeated Friedel–Crafts alkylation. The normal strategy to increase cavity size involves increasing the number of subunits along the ring, but this can concomitantly lead to an increase in conformational flexibility accompanied by a collapse of the cavity. In contrast, the cavity size of biphenarenes is increased by incorporating spacers (or functional modules) between the terminal aromatic units of the building blocks, making macrocycles with large cavity easily accessible. Because the structural units of biphenarenes are independent of each other, the cavity sizes can be easily expanded using long and rigid structural units without affecting the cyclization reaction. For example, Li and co-workers synthesized macrocyclic hosts of terphen[*n*]arenes (TP*n*s, *n* = 3–6) and quaterphen[*n*]arenes (QP*n*s, *n* = 3–6) [61]. TP*n*s and QP*n*s have larger cavities and better self-assembly properties compared to traditional macrocycles because of their longer and more rigid structural units (Table 2). Among them, the largest macrocyclic molecule QP6 has a cavity size of more than 30 Å, which is much larger than that of classic macrocyclic hosts. The customizable cavity sizes of biphenarenes facilitate the encapsulation of large guest molecules (such as polypeptides or other biomacromolecules) and effectively enrich their host–guest properties. Due to these advantages, biphenarenes have potential applications in the field of supramolecular self-assembly.

**Table 1 molecules-28-04422-t001:** Chemical structures and diameters of traditional macrocyclic hosts.

Macrocyclic Host	Chemical Structure	Diameter (Å)	Ref.
*α*-Cyclodextrin	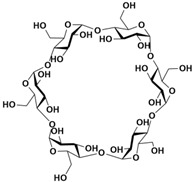	4.7–5.3	[11]
*β*-Cyclodextrin	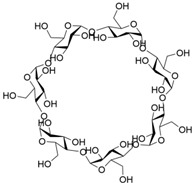	6.0–6.5	[12]
*γ*-Cyclodextrin	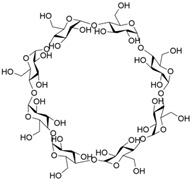	7.5–8.3	[14]
Per-ethylated pillar[5]arene	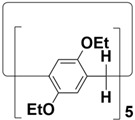	4.7	[4]
Per-ethylated pillar[6]arene	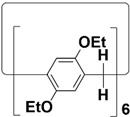	6.7	[62]
Cucurbit[6]uril	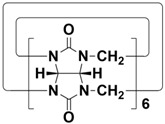	3.9	[56]
Cucurbit[7]uril	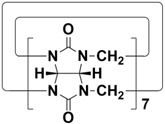	5.4	[56]
Cucurbit[8]uril	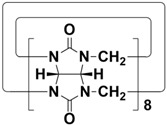	6.9	[56]

(ii) Easy functionalization. Furthermore, the development of new applications of macrocyclic hosts is inseparable from the functionalization of macrocycles [62,63,64]. Whereas the skeletons of most macrocycles cannot be changed, their functional substituents can be introduced on their portals [65,66,67]. The synthesis of biphenarenes is achieved through modular synthetic strategy, and the functionalization can not only introduce functional substituents on their portals, but also realize functionalization by changing the functional modules. By changing the functional modules of biphenarenes by modular synthesis, endo-functionalized macrocycles can be easily synthesized (Figure 2) [47]. This method not only expands the cavity sizes of biphenarenes, but also enriches their host–guest properties and functions. In this way, phenyl, naphthyl, benzofuranyl, benzothiophyl and even π-electron-rich donors (e.g., anthryl, pyrenyl, azophenyl) can also be introduced into the macrocyclic backbones of biphenarenes.

The structural properties of biphenarenes make the synthesis of functional macrocycles more economical and efficient. It is possible to integrate different functional building units into a macrocycle, greatly expanding the toolbox of biphenarenes. The diversity of structures and functions of biphenarenes show extensive binding abilities and extraordinary self-assembly behaviors, laying the foundation for their vigorous development in the field of supramolecular chemistry.

## 3. Molecular Recognition

Molecular recognition plays an important role in biological systems, ion detection, environmental pollution control, etc. [68,69,70,71]. Macrocyclic hosts are widely used in molecular recognition because of their high affinity and selectivity for cationic or neutral guests [72,73,74,75]. Compared with traditional macrocyclic hosts, the unique structure and easy functionalization of biphenarenes give them excellent abilities to selectively bind various types of guests [76,77,78]. These features provide a useful platform for the construction of interesting supramolecular systems. In addition, biphenarenes are easy to obtain and functionalize, which provides flexibility for building efficient recognition systems, and is expected to become popular macrocyclic hosts in the future.

Since most biological functions and processes occur in aqueous media, molecular recognition in water is extremely important [71]. Li and co-workers designed and synthesized anionic water-soluble biphen[3]arenes (H1) and investigated their host–guest complexation with a series of cationic guests of different sizes and shapes (G1–G10) (Figure 3A) [78]. The cleavage of the ether groups in perethylated biphen[3]arene and perethylated biphen[4]arene by reaction with excess BBr_3_ in CH_2_Cl_2_ could quantitatively produce H2 and H3, respectively. The binding strength of H1-H3 to these guests was quantitatively estimated by ^1^H NMR titration (Table 3). The ^1^H NMR spectra showed that all protons of G2 were shielded in the presence of H1, indicating that H1 formed a host–guest inclusion complex with G2 (Figure 3B). At the same time, the color change of the solution before and after the complexation also indicated the occurrence of the host–guest complexation (Figure 3C). This phenomenon also made H1 a good material for the detection of G2, indicating the potential application of H1 in the detection of guest molecules such as paraquat.

Huang, Yu and co-workers studied the molecular recognition of a series of water-soluble biphenarenes [77,79]. They found that G11 could act as an axis through the cavity of H1, forming a 1:1 complex (Figure 4A) [77]. The existence of host–guest complexation was confirmed by ^1^H NMR (Figure 4B). Interestingly, under the appropriate molar ratio, H1 and G12 showed an obvious Tyndall effect in aqueous solution, and there were abundant aggregates (Figure 4C). By adjusting the pH value, the conversion between micelles formed by G12 and vesicles based on G12⸦H1 was realized. Subsequently, they also investigated the host–guest complexation of cationic water-soluble biphen[3]arene with sodium 1-hexanesulfonate, and further used it to alter the aggregation behavior of amphiphilic guests in water [79]. In addition, they synthesized [2]calix[1]biphenyl-type hybrid[3]arene with a biphenyl unit, showing good complexation with 1-dihexylammonium hexafluorophosphate [80]. Recently, they constructed the first nonporous adaptive crystals of [2]calix[1]biphenyl-type hybrid[3]arene for the efficient separation of benzene and cyclohexane [81].

The electron-rich cavities of biphenarenes have a strong affinity for various cationic and electron-deficient neutral guests, and show excellent molecular recognition abilities. This renders them useful as sensors, nanomaterials, ion or molecular transporters and supramolecular amphiphiles. In particular, research on water-soluble biphenarenes will help to understand and model biological processes. Because of the easy modification, water-soluble biphenarenes have been widely studied and have shown great potential in molecular recognition.

## 4. Adsorption and Separation

Adsorption and separation are fundamental technologies in many industrial fields [82]. They can be used to efficiently extract, concentrate and refine compounds, and play a role in separation and purification in the production process [83,84,85,86,87,88,89]. Adsorption and separation are widely used in the fields of environmental protection [89,90], the chemical industry [36,91,92], water treatment [93] and nuclear waste concentration [94,95,96]. As important chemical raw materials, hydrocarbons are separated and purified mainly by distillation and fractionation in traditional petrochemical production. However, traditional separation technologies suffer from high costs and low efficiency, while accounting for a large part of the world’s energy consumption [86].

In recent years, a variety of porous materials have been used for adsorption and separation in order to develop economical and efficient adsorption and separation technologies. For instance, zeolites [97,98], metal–organic frameworks (MOFs) [98,99,100,101,102,103] and covalent organic frameworks (COFs) [104,105] have rigid structures, high specific surface areas and regular adjustable pores. They are widely used in the fields of adsorption and separation, environmental improvement, industrial production and biomedicine. However, these porous materials also have drawbacks that cannot be ignored. The structures of MOFs are easily destroyed in high temperature or acid–base environments, and their chemical stability is weak. Furthermore, the rigid structures of these materials lead to poor solubility and lack of solution processing properties [106,107].

Macrocyclic hosts such as crown ethers, cyclodextrins, calixarenes, cucurbiturils and pillararenes are widely used for adsorption and separation of gases, organic pollutants, nuclear wastes and hydrocarbons [108,109,110,111,112,113]. With the rapid development of macrocyclic hosts, excellent results have been obtained in the field of adsorption and separation of hydrocarbons [114,115,116,117]. For example, Huang and co-workers reported the application of nonporous adaptive crystal (NAC) materials based on pillararenes in adsorptive separation of important hydrocarbons and nuclear waste pollution [118,119]. Compared with traditional porous materials, NACs are nonporous in the initial crystalline state, but transformations of intrinsic or extrinsic pores along with crystal structures are induced by preferential guest molecules [120,121]. More importantly, NACs are able to return to their original nonporous structures after the removal of the guest molecules. Macrocycle-based NAC materials are uniquely attractive in the field of hydrocarbon adsorption and separation because of their excellent adsorption and separation properties, recyclability and stability.

As a new generation of macrocyclic hosts, biphenarenes also play an indispensable role in adsorption and separation. In 2016, Yang and co-workers first synthesized [2]biphenyl-extended-pillar[6]arenes ([2]Bp-ExP6) with rigid nanocavities [122]. *m*-xylene and toluene could be efficiently bound in the cavity of [2]Bp-ExP6, indicating that [2]Bp-ExP6 has great potential in the purification of hydrocarbons. Later, Li and co-workers efficiently synthesized 2,2′,4,4′-biphen[3]arenes (H4) for the separation of *cis-* and *trans*-1,2-dichloroethylene (*cis*-DCE and *trans*-DCE) isomers (Figure 5A) [44]. They found that H4*α* exhibited selective adsorption to *cis*-DCE (Figure 5B). The time-dependent solid–vapor adsorption of H4*α* with *cis-*/*trans*-DCE mixtures showed that the uptake of *cis*-DCE was very rapid, almost reaching saturation after only 10 min. At the same time, H4*α* showed excellent selectivity for *cis-*/*trans*-DCE mixtures. The adsorption capacity of H4*α* for *cis*-DCE was close to 0.7 equivalent, while that for *trans*-DCE was only 0.05 equivalent (Figure 5C). Repeatedly performing the experiment showed that the adsorption selectivity and adsorption capacity of H4*α* toward *cis*-DCE did not change significantly (Figure 5D). Activated H4*α* crystals had good separation efficiency and recyclability. The simple synthesis, excellent selectivity and recyclability of biphenarenes make them be one of the most promising NAC materials.

In order to meet the growing demand for energy and mitigate the greenhouse effect, scientists are stepping up research into clean and efficient nuclear energy [123,124]. However, the contamination of nuclear waste is a major challenge facing the development of nuclear energy. One is radioactive iodine, a volatile contaminant of nuclear waste. ^129^I has a particularly long half-life (~10^7^ years) and is a dangerous isotope of iodine. ^131^I has a short half-life (8.02 days), but is highly volatile and can interfere with metabolic processes in the human body. Therefore, it is very important to develop materials that capture highly volatile iodine. Li and co-workers designed and synthesized H5-H12 with different cavity sizes (Figure 6A) [61]. Interestingly, cyclic pentamers and hexamers (H7, TP6, QP5 and QP6) could easily form supramolecular organogels (G-H7, H8, G-H11 and G-H12) in dichloromethane/hexane solutions. Among them, G-H7 could be used as an excellent material for iodine capture, not only effectively adsorbing iodine molecules in water, but also efficiently capturing iodine vapor (Figure 6B). When G-H7 was exposed to iodine vapor, the adsorption amount of iodine increased gradually with time, indicating that G-H7 had a good adsorption ability for iodine vapor (Figure 6C). After exposure to iodine vapor, the color of the xerogel gradually changed from white to almost black. At the same time, approximately 92% of the iodine in G-H7 was rapidly absorbed within 30 min (Figure 6D). In addition, G-H7 had high recyclability and chemical stability (Figure 6E). Therefore, H5-H12 hold broad application prospects in pollutant sequestration.

In addition, many biphenarene analogues also show excellent adsorption and separation properties. Yang and co-workers obtained an elongated-geminiarene by replacing the xylylene units with biphenylene units [125]. With large-scale synthesis, a nano-sized cavity and excellent solid-state host–guest properties, elongated-geminiarene combines the advantages of both geminiarenes and biphenarenes. The elongated-geminiarene-based crystalline materials showed high efficiency in the separation of aromatic and cyclic aliphatic compounds. They also found that [2]biphenyl-extended pillar[6]arene derivatives could effectively separate carbon dioxide [126]. In addition, Huang and co-workers reported the synthesis of hybrid[3]arene by 4,4′-biphenol diethyl ether and 1,3,5-trimethoxybenzene [118]. With the hybrid[3]arene-based NAC materials, benzene can be completely separated from a mixture of benzene and cyclohexane.

Biphenarenes show a great development scope in the field of adsorption and separation due to the advantages of simple synthesis, stable structure and excellent properties. In addition, biphenarenes with large cavities show great potential for adsorbing and separating large sizes of guests.

## 5. Drug Delivery

Drug delivery has enabled the development of many drug products [127,128,129]. Drug delivery can enhance the delivery of drugs to target cells and minimize off-target effects. As therapeutics evolve from small molecules to nucleic acids, peptides, proteins and monoclonal antibodies, drug delivery also faces new challenges.

Macrocyclic hosts have gained remarkable achievements in the field of drug delivery [130,131]. The non-covalent interactions between various hosts and guests enable highly adjustable combinations and intelligent stimuli-response properties. The excellent stimuli-responsive properties enable macrocyclic hosts to trigger drug release in response to pH, light, chemical or electrochemical stimuli [132,133,134,135]. To meet the needs of supramolecular nanomedicine, the cavities of macrocyclic hosts must be large enough to accommodate various drug–drug coupling molecules. However, limited by the cavity size, traditional macrocyclic hosts are excellent molecular containers for small- or medium-sized guests, but cannot accommodate biomacromolecules. Biphenarenes can be modularized by the selection of long and rigid structural units to obtain macrocycles with large cavities. As the cavity size of macrocyclic hosts increases, large guest molecules can be encapsulated, which effectively expands the application of macrocyclic hosts in drug delivery [136,137].

Li and co-workers found that two water-soluble quaterphen[*n*]arenes (H13, H14) with large-sized cavities and interesting host–guest properties were able to achieve an overall complexation towards peptides (Figure 7A) [138]. The host–guest inclusion modes significantly inhibited the hemolytic toxicity of pexiganan (PXG) in rabbit red blood cells (rRBCs) and improved its metabolic stability without affecting the antibacterial activity (Figure 7B). When the concentration of PXG was 160 mM, the hemolysis rate of rRBCs taking up PXG/H13 (80.51 ± 2.83%) was about 20% lower than that of rRBCs without H13. More importantly, H14 significantly reduced the hemolysis rate of rRBC to 26.94 ± 0.96% (Figure 7C,D), indicating that the strong complexation of H14 effectively blocked the reaction of PXG with rRBCs. This typical example illustrates the potential application of biphenarenes in the encapsulation of biological macromolecules such as peptides/proteins.

Li and co-workers further synthesized a water-soluble quaterphen[*n*]arenes bearing dendritic multicarboxylate moieties (H15) (Figure 8A) [139]. Compound H15 was found to have a high binding affinity for LyeTxΙ (LyeTxΙ is a high-molecular-weight biotoxin isolated from *Lycosa erythrognatha* spider venom) and its association constant (*K*a) was (7.01 ± 0.18) × 10^7^ M^−1^. They found that the cytotoxicity of H15 to cells was negligible (Figure 8B). On the contrary, the cytotoxicity of LyeTxI was significantly reduced in the presence of H15 (Figure 8C). Host–guest complexation of LyeTxI and H15 protected the cell membrane by destroying LyeTxI and significantly inhibited cytotoxicity and hemolysis of erythrocytes (Figure 8D–F). In addition, the survival rate of LyetxI-poisoned mice was improved by emergency administration of H15. This result indicates that a supramolecular strategy based on host–guest complexation by large-sized macrocycles is expected to be a general method to detoxify macromolecular biotoxins.

Recently, Li and co-workers designed and synthesized quaternary ammonium per-functionalized biphen[*n*]arenes (H16, H17) with excellent biofilm resistance (Figure 9A) [140]. Compared with phosphate-buffered saline, cefazolin sodium (CFZ)/H16 and CFZ/H17 could effectively disrupt cell colonies in mature *E. coli* biofilms and reduce the bacterial density (Figure 9B). Furthermore, administration of CFZ/H16 or CFZ/H17 complexes induced significant destruction of biofilms and a sharp reduction in *E. coli* bacteria (Figure 9C). At the same time, the surface morphology of *E. coli* exposed to free CFZ showed membrane lysis, and the complex of H16 or H17 retained the antibacterial effect of CFZ (Figure 9D). This study demonstrated that biphenarene-based disruptors can effectively preserve the broad-spectrum sterilizing effect of antibiotic agents.

In addition, a series of novel macrocyclic hosts named biphenyl-extended pillararenes have been designed by Yang and co-workers [44,141]. These macrocycles have rigid skeleton structures and multifunctional modification sites similar to pillararenes. Biphenyl-extended pillararenes have the advantages of an electron-rich cavity structure with extended size and high synthesis yield due to the introduction of biphenyl units. Yao and co-workers successfully synthesized anionic water-soluble [2]biphenyl-extended-pillar[6]arene (H18) (Figure 10A) [142]. Based on the host−guest molecular recognition between H18 and chlorambucil (CB), a supramolecular nanoprodrug (SNP) was fabricated using H18 and the drug−drug conjugate guest (IR806-CB). In the acidic tumor cell microenvironment with a high concentration of glutathione (GSH), SNP could be rapidly degraded and efficiently released to activate CB through the GSH cleavage of the disulfide linker. Under the near-infrared irradiation, IR806 produced hyperthermia and ROS to kill tumor cells. Near-infrared fluorescence imaging showed that the SNP nanoprodrug had excellent tumor aggregation and drug retention (Figure 10B). The hyperthermia study further demonstrated that SNP had significant photothermal conversion efficiency and tumor-targeting ability (Figure 10C). By monitoring the tumor size, it was found that both IR806 + NIR and SNP groups had significant inhibitory effects on tumor growth (Figure 10D–F). These results proved that SNP formed by H18 and IR806-CB could significantly improve therapeutic efficiency through synergistic photodynamic therapy, photothermal therapy and chemotherapy.

Overall, the fascinating structures and large cavities of biphenarenes lay the foundation for drug delivery and cancer therapy.

## 6. Fluorescence Sensing

Fluorescence sensors are widely used to detect various analytes because of their high sensitivity, strong specificity and fast response speed [143]. Organic luminescent materials with high quantum efficiency have attracted wide attention due to their applications in sensors, bioimaging, laser displays, light-emitting diodes and anti-counterfeiting technologies [144,145,146,147]. It is important to design organic luminescent materials with a simple molecular structure, adjustable properties and excellent thermal stability for the construction of high-performance fluorescence sensors. However, most organic luminescent materials suffer from quenching effects in the aggregate state due to the formation of detrimental aggregates such as excimers and exciplexes. This severely limits their applications in fluorescence sensing and organic luminescent materials. In contrast, aggregation-induced emission (AIE) molecules are non-emissive in dilute solutions, but their luminescence is significantly enhanced when the molecules are aggregated [148]. AIE materials with high solid-state luminescence efficiency are expected to fundamentally solve the ACQ problem of traditional organic luminescent materials [149,150]. Therefore, there is an urgent requirement to develop a new fluorescence enhancement strategy. This not only contributes to the construction of excellent fluorophores and materials, but also plays an important role in understanding the relationship between luminescence mechanisms and molecular structures.

Based on this, Li and co-workers proposed an effective and universal strategy for enhancing solid-state emission of luminophores, known as macrocyclization-induced emission enhancement (MIEE) [151]. A benzothiadiazole-based macrocycle (H19) with three methylene bridges was obtained (Figure 11A). H19 exhibited redshift emission compared to BT-M (Figure 11B). In addition, the photoluminescence spectra showed that H19 had a higher solid state fluorescence quantum yield than BT-M. MIEE not only effectively improved the fluorescence efficiency of organic luminophores, but also has good universality, which is conducive to the development of organic luminophores.

Recently, Li and co-workers developed a synthetic strategy for heterogeneous macrocycles that aimed to integrate different functional groups into one macrocyclic backbone [152]. They successfully obtained isomeric macrocycles containing functional groups of fluorenone and fluorenol by both post-modification and one-pot methods (Figure 12A). Notably, the photophysical characterization revealed that the fluorenol was part of the energy donor and the fluorenone was part of the energy acceptor (Figure 12B). Macrocycles H20, H21 and H22 showed similar photoluminescence spectra with the same yellow emission peak at 554 nm which was assigned to the emission of the fluorenone moiety, while H23 had a blue emission peak at 393 nm (Figure 12C). Moreover, the emission intensity of the fluorenone moiety followed the order of H22 > H21 > H20 (Figure 12D). Meanwhile, intramolecular energy transfer had no effect on the radiation attenuation of fluorenone (Figure 12E). The obtained isomeric macrocycles showed interesting intramolecular energy transfer and fluorescence enhancement due to well-matched absorption/emission spectra and the close distance between the energy donor and the acceptor. This work provides a new method for the efficient synthesis of multiphase functional macrocycles.

It is not difficult to see that the modular synthetic strategy of biphenarenes is beneficial to the development of functional macrocyclic hosts with different properties. In addition, by integrating different structural units into one macrocycle, more interesting structures and applications can be developed and explored.

## 7. Conclusions

In conclusion, the modular synthetic strategy gives biphenarenes rich functions and host–guest properties. Compared with traditional macrocyclic hosts, biphenarenes have unique advantages such as customizable cavity size, diverse skeletons and alternative binding sites. As a kind of “young” macrocycle, biphenarenes have shown their brilliance in the fields of supramolecular chemistry, and will have broad development space in the future. Through the modular synthetic strategy, different building units can be used to obtain macrocyclic hosts with large cavities. This opens up a broad perspective for the complexation of macromolecules. At present, biphenarenes play important roles in drug delivery, and cancer diagnosis and treatment. Furthermore, since biphenarenes can be easily functionalized, supramolecular functional materials can be constructed by introducing functional modules or post-modification methods. The structures and properties of biphenarenes are expected to be expanded greatly by the reasonable design of diverse skeletons. We hope that this review will deepen the interest in supramolecular macrocyclic chemistry and stimulate the research on biphenarenes and synthetic macrocyclic hosts.

## Figures and Tables

**Figure 1 molecules-28-04422-f001:**
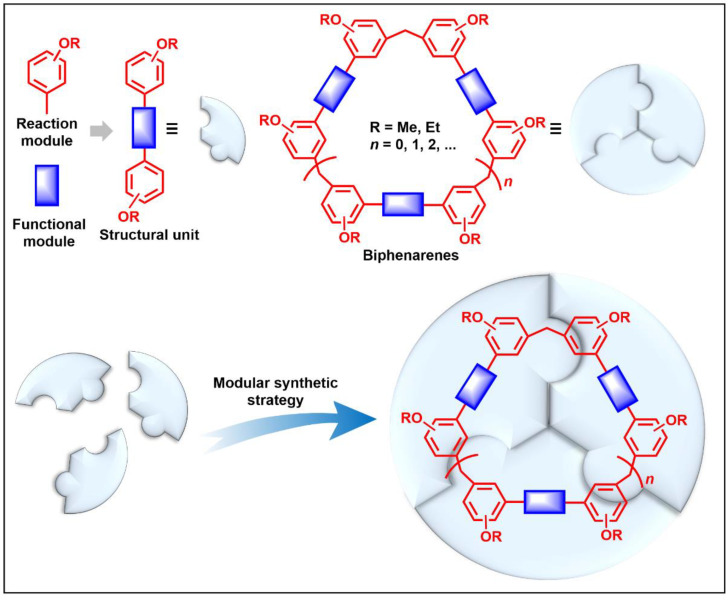
Modular synthetic strategy of biphenarenes.

**Figure 2 molecules-28-04422-f002:**
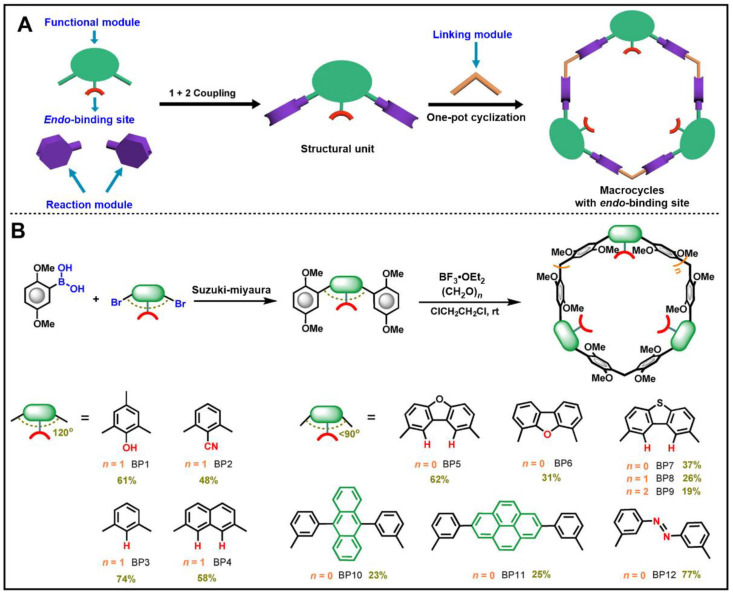
(**A**) Cartoon representation of modular introduction of endo-binding sites in macrocyclic cavity. (**B**) Modular synthesis of macrocycles with customizable endo-binding sites. Reproduced from ref. [47] with permission from John Wiley and Sons.

**Figure 3 molecules-28-04422-f003:**
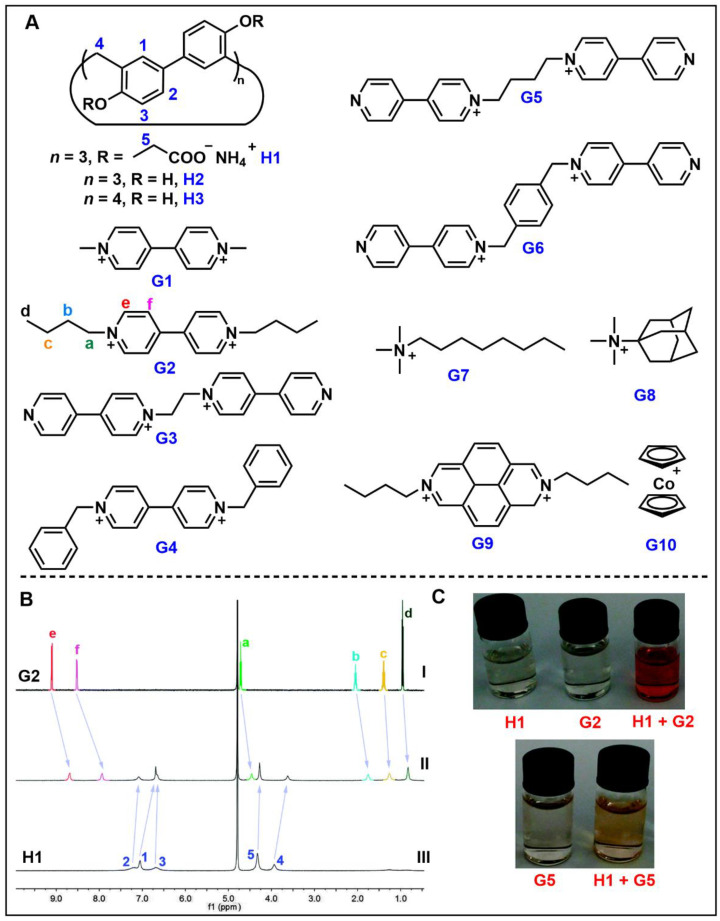
(**A**) Structures of biphenarenes and cationic guests. (**B**) ^1^H NMR spectra of (I) G2, (II) G2 + H1 and (III) H1. (**C**) Color changes of G2 and G5 after complexing with H1. Reproduced from ref. [78] with permission from the Royal Society of Chemistry.

**Figure 4 molecules-28-04422-f004:**
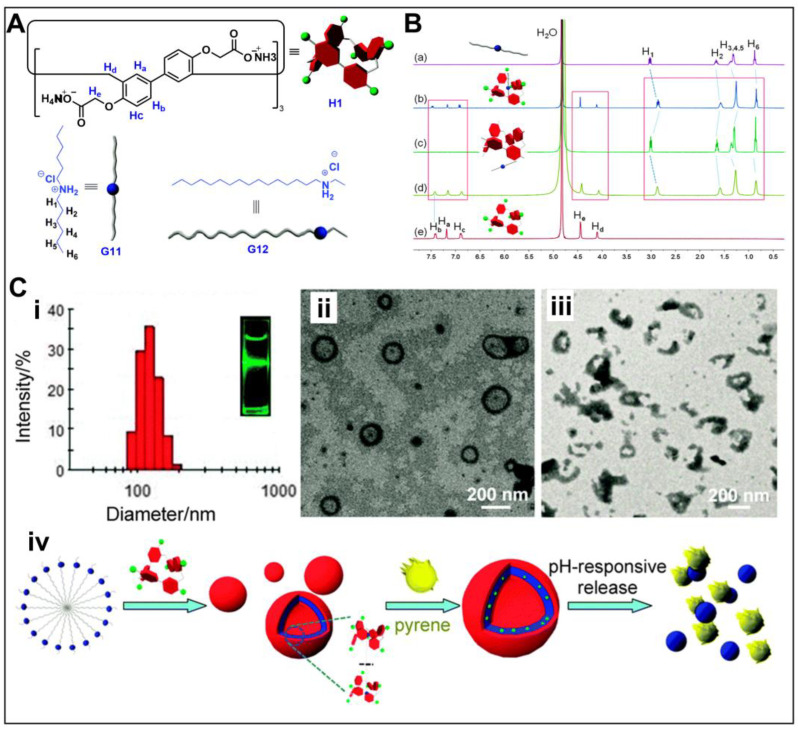
(**A**) Structures of H1, G11 and G12. (**B**) ^1^H NMR spectra: (a) G11; (b) H1 and G11; (c) after addition of 2 µL of aqueous DCl solution (35 wt%) to (b); (d) after addition of 3.5 µL of aqueous NaOD solution (30 wt%) to (c); (e) H1 (1.00 mM) (Red boxes, ^1^H NMR spectra change of H1 before and after addition of DCl and NAOD). (**C**) Self-assembling behaviors of H1⸧G12 (i) DLS data of H1⸧G12 aggregates; TEM images: (ii) H1⸧G12 aggregates; (iii) H1⸧G12 aggregates after the solution pH was adjusted to 4.0; (iv) illustration of the formation of the aggregates and the process of pH-responsive release of pyrene molecules. Reproduced from ref. [77] with permission from the Royal Society of Chemistry.

**Figure 5 molecules-28-04422-f005:**
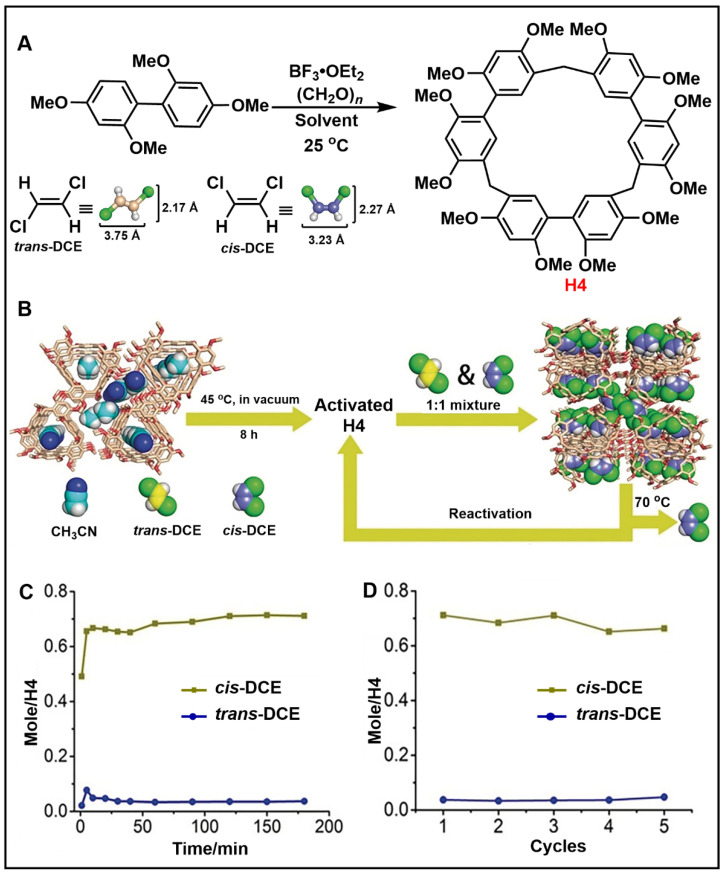
(**A**) Synthesis of H4 and structures of *cis*-DCE and *trans*-DCE. (**B**) Schematic representation of purification of the DCE isomer by H4. (**C**) Time-dependent solid–vapor adsorption plots of H4*α*. (**D**) Cyclic adsorption capacities of H4*α*. Reproduced from ref. [44] with permission from John Wiley and Sons.

**Figure 6 molecules-28-04422-f006:**
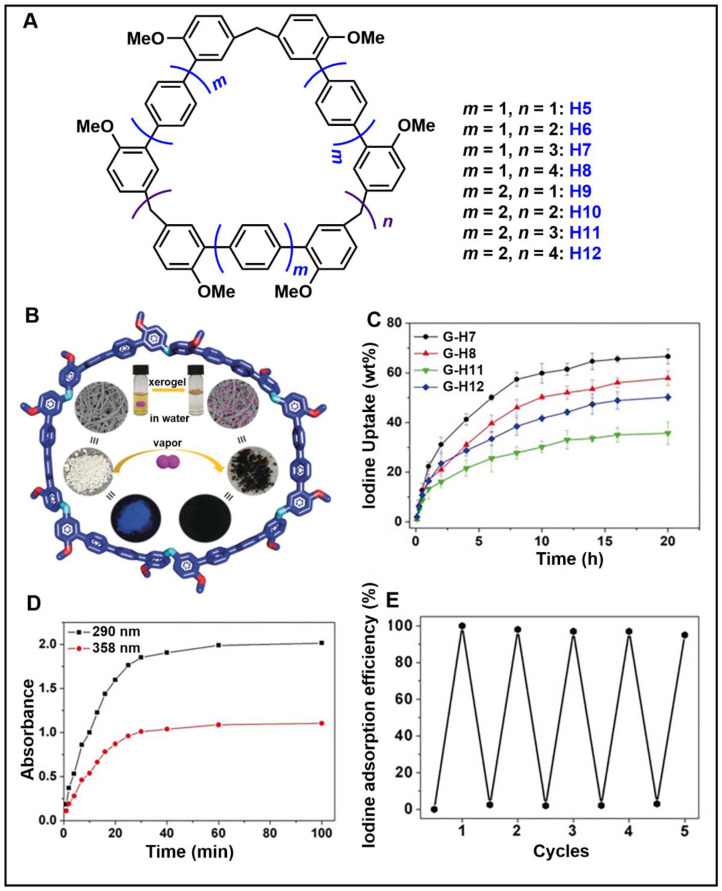
(**A**) Structures of H5-H12. (**B**) Schematic representation of the iodine capture process. (**C**) Time-dependent iodine uptake by xerogels. (**D**) Top: color changes of G-H7 after adsorption of iodine; bottom: fluorescence images of G-H7 and iodine@G-H7 under a UV lamp (365 nm). (**E**) Cyclic adsorption and desorption efficiencies of G-H7 for iodine. Reproduced from ref. [61] with permission from John Wiley and Sons.

**Figure 7 molecules-28-04422-f007:**
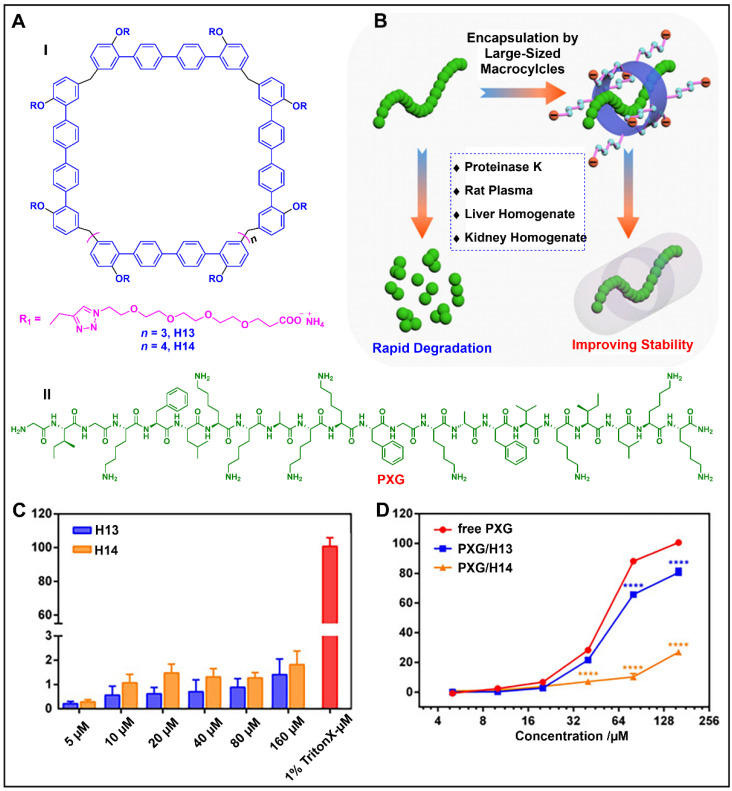
(**A**) Structure of quaterphen[*n*]arenes (H13, H14) (I) and PXG (II). (**B**) Schematic representation of H13/H14 improving the metabolic stability of a peptide. (**C**) Hemolysis of H13 and H14 toward rRBCs, rRBCs incubated with 1% Triton X-100 were used as a positive control. (**D**) Cytotoxicity of free PXG and PXG complex, ^☆☆☆☆^ *p* < 0.0001. Reproduced from ref. [138] with permission from John Wiley and Sons.

**Figure 8 molecules-28-04422-f008:**
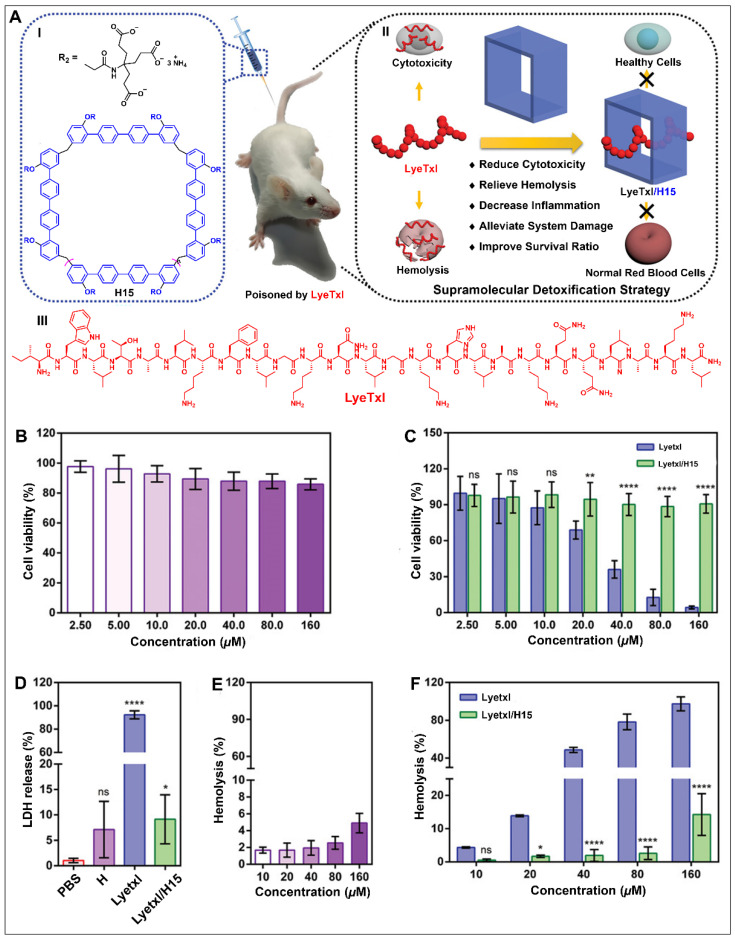
(**A**) Structure of quaterphen[*n*]arenes (H15) (I), schematic illustration of H15 as a supramolecular antidote against LyeTxI (II) and structure of LyeTxI (III). (**B**) Viability of human normal renal epithelial cells (293T) cells treated with different concentrations of H15. (**C**) Viability of 293T cells administered with different concentrations of free LyeTxI and LyeTxI/H15. (**D**) LDH release of 293T cells treated with LyeTxI in the absence and presence of H15. (**E**) Hemolysis of H15 toward rRBCs. (**F**) Hemolytic activity of free LyeTxI and LyeTxI/H15. ns, not significant. ^☆^
*p* < 0.05, ^☆☆^
*p* < 0.01, ^☆☆☆☆^
*p* < 0.0001. Reproduced from ref. [139] with permission from John Wiley and Sons.

**Figure 9 molecules-28-04422-f009:**
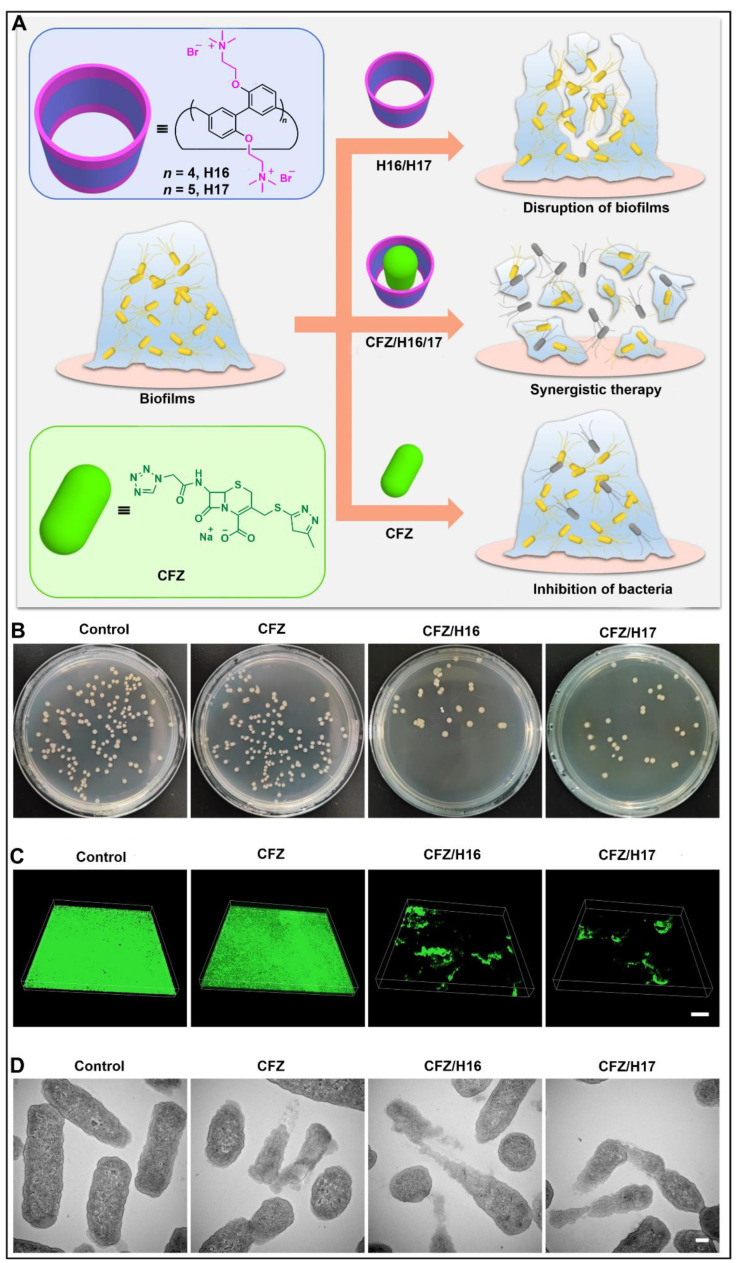
(**A**) Structures of H16, H17 and CFZ, and schematic illustration of supramolecular synergistic antibacterial strategy. (**B**) Images of colony-forming units of *E. coli* after different treatments. (**C**) CLSM 3D images of *E. coli* biofilms after different treatments. (**D**) TEM images of ultrathin sections of *E. coli* after different treatments. Reproduced from ref. [140] with permission from John Wiley and Sons.

**Figure 10 molecules-28-04422-f010:**
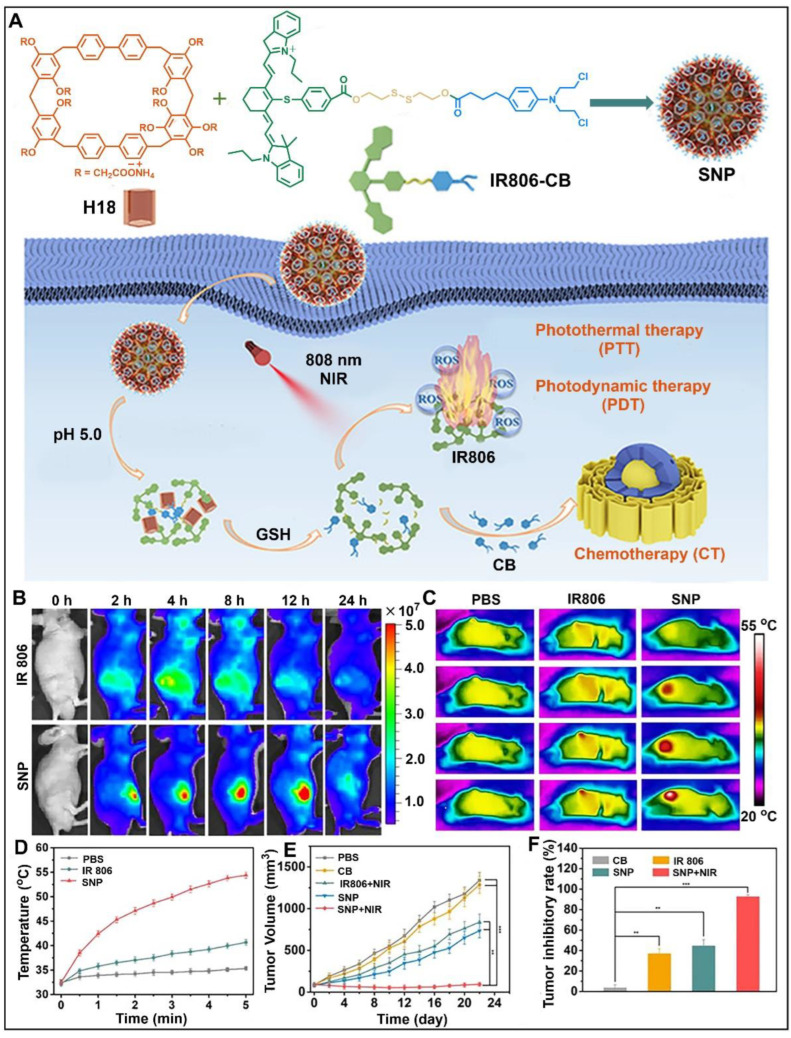
(**A**) Schematic illustration of the fabrication of supramolecular nanoprodrug SNP and the mechanism of PDT-PTT-CT combination therapy. (**B**) In vivo fluorescent imaging of HeLa tumor-bearing mice after intravenous injection of IR806 and SNP at selected time intervals. (**C**) Near-infrared thermal images and (**D**) temperature profiles of mice treated with free IR806 or SNP under NIR irradiation. (**E**) Tumor volume changes and (**F**) tumor inhibition rates for different groups. ^☆☆^
*p* < 0.01, ^☆☆☆^
*p* < 0.001. Reproduced from ref. [142] with permission from the American Chemical Society.

**Figure 11 molecules-28-04422-f011:**
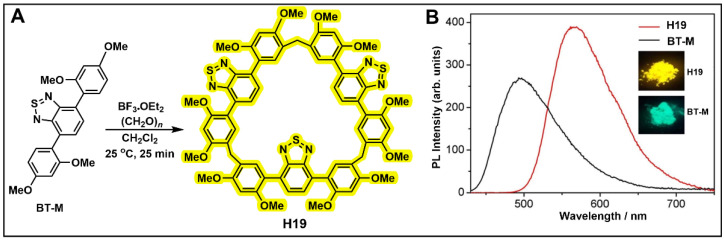
(**A**) The synthetic route of H19. (**B**) Photoluminescence spectra of BT m and H19 in the solid state. (reproduced with permission of Springer Nature from ref. [151]).

**Figure 12 molecules-28-04422-f012:**
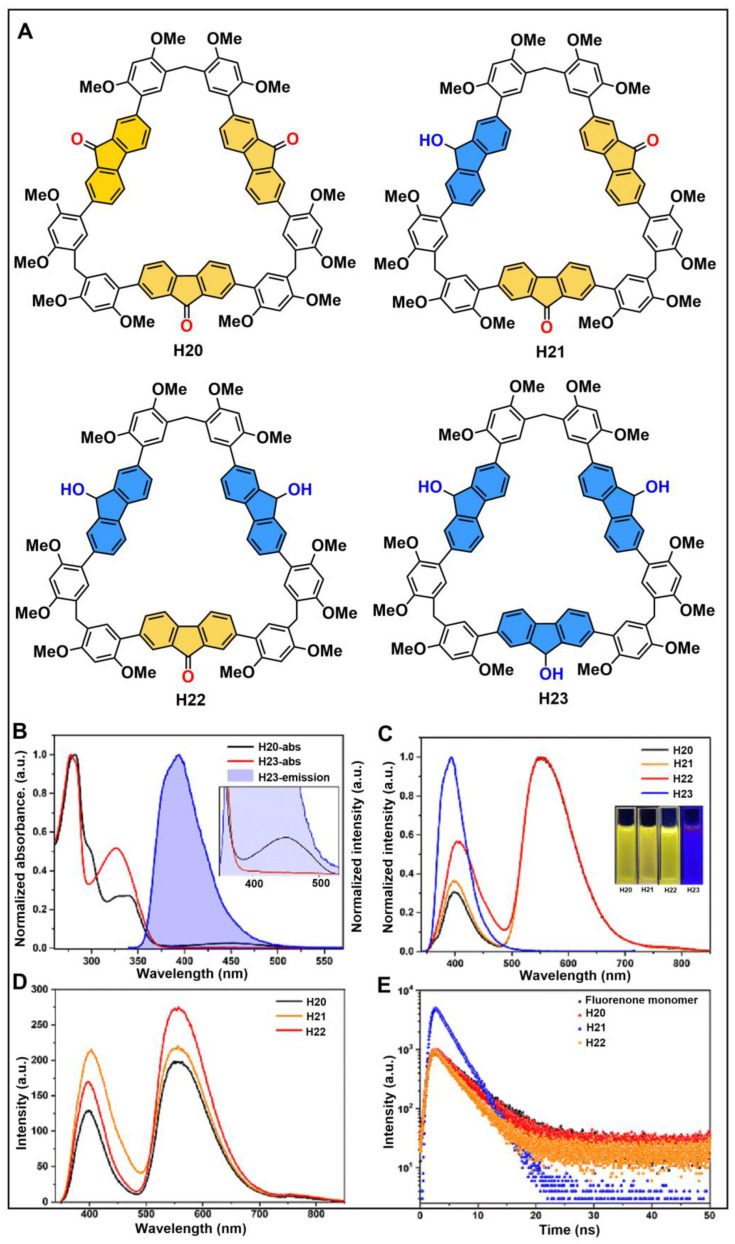
(**A**) Structures of heterogeneous macrocycles. (**B**) UV-*vis* spectra of H20, H23 and photoluminescence spectra of H3. (**C**) Normalized photoluminescence spectra of H20, H21, H22 and H23. (**D**) Photoluminescence spectra of H20, H21 and H22. (**E**) Time-resolved PL decay of fluorenone monomer, H20, H21 and H22. Reproduced from ref. [152] with permission from the Royal Society of Chemistry.

**Table 2 molecules-28-04422-t002:** Chemical structures and diameters of typical biphenarenes.

Macrocyclic Host	Chemical Structure	Diameter (Å)	Ref.
Terphen[3]arene	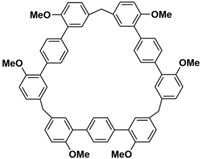	9.86	[61]
Terphen[4]arene	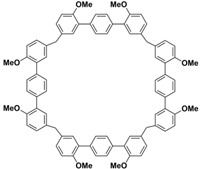	13.85	[61]
Terphen[5]arene	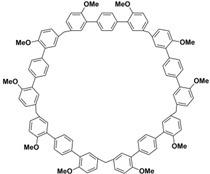	16.47	[49]
Terphen[6]arene	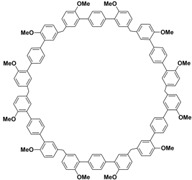	21.59	[49]
Quaterphen[3]arene	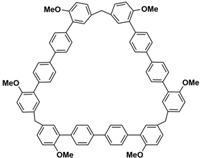	15.76	[61]
Quaterphen[4]arene	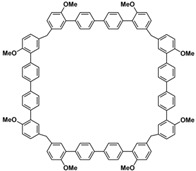	17.86	[61]
Quaterphen[5]arene	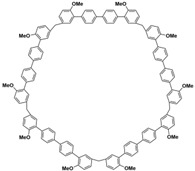	21.48	[61]
Quaterphen[6]arene	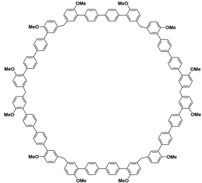	30.82	[61]

**Table 3 molecules-28-04422-t003:** Association constants (*K*_a_) for the host–guest complexes of biphenarenes with different guests.

Host	Guest	Solvent	*K*_a_ (M^−1^)
H1	G1	D_2_O	(1.1 ± 0.2) × 10^4^
H1	G2	D_2_O	(2.4 ± 0.1) × 10^4^
H1	G3	D_2_O	(5.1 ± 0.3) × 10^4^
H1	G4	D_2_O	(4.7 ± 0.4) × 10^3^
H1	G5	D_2_O	(9.6 ± 1.7) × 10^3^
H1	G6	D_2_O	(1.5 ± 0.2) × 10^3^
H3	G7	acetone-*d*_6_	(0.32 ± 0.04) × 10^2^
H3	G8	acetone-*d*_6_	(1.6 ± 0.2) × 10^2^
H3	G9	acetonitrile-*d*_3_	(2.4 ± 0.3) × 10^2^
H3	G10	acetone-*d*_6_/CD_2_Cl_2_ (1:1, *v*/*v*)	(3.1 ± 0.3) × 10^3^

## Data Availability

Not applicable.

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
